# Effects in Sleep and Recovery Processes of NESA Neuromodulation Technique Application in Young Professional Basketball Players: A Preliminary Study

**Raquel Medina-Ramírez** [1], **Milos Mallol Soler** [2], **Franc García** [2], **Francesc Pla** [2], **Aníbal Báez-Suárez** [1], **Esther Teruel Hernández** [1,*], **D. David Álamo-Arce** [1] **and María del Pino Quintana-Montesdeoca** [1]

1   Faculty of Health Sciences, University of Las Palmas de Gran Canaria, 35048 Las Palmas de Gran Canaria, Spain; raquel.medina@ulpgc.es (R.M.-R.); anibal.baez@ulpgc.es (A.B.-S.); danieldavid.alamo@ulpgc.es (D.D.Á.-A.); mariadelpino.quintana@ulpgc.es (M.d.P.Q.-M.)

2   Science Hub Barcelona, 08001 Barcelona, Spain; milos.mallol@fcbarcelona.cat (M.M.S.); francgarciagarrido@gmail.com (F.G.); francesc.pla@fcbarcelona.cat (F.P.)

*   Correspondence: esther.teruel@carm.es

**Abstract:** The competitive calendars in sports often lead to fluctuations in the effort-recovery cycle and sleep quality. NESA noninvasive neuromodulation, achieved through microcurrent modulation of the autonomic nervous system, holds promise for enhancing sleep quality and autonomic activation during stressful situations. The objective of this study was to analyze the sleep and recovery responses of basketball players over six weeks of training and competition, with the integration of NESA noninvasive neuromodulation. A preliminary experimental study involving 12 participants was conducted, with a placebo group (*n* = 6) and an intervention group (*n* = 6) treated with NESA noninvasive neuromodulation. Sleep variables and biomarkers such as testosterone, cortisol, and the cortisol:testosterone ratio were analyzed to assess player recovery and adaptations. Significant differences were observed in total, duration, and REM sleep variables (*p*-value= < 0.001; 0.007; <0.001, respectively) between the intervention and placebo groups. The intervention group demonstrated increased duration of sleep variables. Cortisol levels showed normalization in the experimental group, particularly in the last two weeks coinciding with the start of playoffs. This study highlights the potential of NESA noninvasive neuromodulation to enhance sleep quality despite challenging circumstances, providing valuable insights into the management of athlete recovery in competitive sports settings.

**Keywords:** performance; team sport; training; sleep; recovery

## 1. Introduction

Insufficient duration and poor quality of sleep have detrimental effects on athletic performance. A bad sleep quality can increase the risk of injury and illness, impair wound healing, diminish sport performance, compromise holistic well-being, and hinder athlete development [1]. Recent studies on sleep quality have revealed heightened levels of sleep complaints among elite team athletes [2]. Additionally, three key risk factors contributing to sleep disturbance have garnered widespread recognition: training, competition, and travel. It is firmly established that sleep disturbances in athletes can be directly linked to fatigue and indirectly influenced by stress and anxiety [1,3].

Recent reviews on sleep and sports have indicated that poor sleep quality may result from strenuous exercise and lengthy training sessions, leading to deteriorated sleep latencies, increased sleep fragmentation, non-restorative sleep, and daytime fatigue [4]. Moreover, the evidence suggests that competition periods, travel, and intensive training are all factors likely to contribute to suboptimal sleep quality among elite teams across various age groups [3].

The ramifications of sleep deprivation are widely recognized, with sleep being a crucial process for several physiological restorative systems. Additionally, recent evidence has underscored the link between insufficient sleep and physiological processes such as carbohydrate metabolism, potentially leading to insulin resistance [5].

Internal load monitoring is defined as a complex data collection process due to the intermittent nature of team sports, in which there are useful and highly utilized variables, such as heart rate. In order to examine the effect of external load in team sports players, a number of researchers have focused on biomarker concentration, with the objective to individualize the analysis of athlete responses [6–8]. Previous research examined levels of cortisol and testosterone hormones and their relationship as a testosterone:cortisol ratio related to the sympathetic and parasympathetic processes during the training load and competitions [9–12].

There is some evidence that sleep treatment intervention can improve the quality and the duration of athletes' sleep, although most of it is based on behavioral-cognitive therapies such as sleep habits [13,14]. Other noninvasive neuromodulation techniques such as TDCs have demonstrated an effect in sporting performance on physical endurance (time to fatigue), physical strength, or visuomotor skill, but not for sleep improvement directly [15]. Cranial electrotherapy stimulation (CES) has been demonstrated to decrease negative emotions and choice reaction times, which can impair sport performance. In recent years, a new kind of noninvasive neuromodulation technique called NESA (Spanish name Neurestimulación superficial aplicada) has been gaining attention in autonomic nervous system treatment and sleep changes [16,17] as a part of daily preventive treatment and training.

The main hypothesis of this preliminary study is that NESA noninvasive neuromodulation (NESA) may be a useful and effective tool in elite sport to improve and optimize sleep quality and the biomarkers related to stress and muscle damage; thus, it could improve the athlete's "neuroefficiency". The objective of this study was to analyze possible differences in sleep variables and cortisol and testosterone concentrations between the intervention and placebo groups during NESA treatment through microcycles.

## 2. Materials and Methods

### 2.1. Study Desig2

This study was a preliminary experimental study with a placebo group. The players' external and internal load data were collected over 6 weeks (6 microcycles) of competition between March and May during the 2020–2021 season. The team concluded a cumulative total of 23 training sessions and participated in nine matches. Players who sustained injuries during games or did not play for a minimum total duration of five minutes were excluded from the study's analysis [18]. The study followed the CONSORT guidelines.

### 2.2. Ethics

All the participants provided a written informed consent prior to being assigned to a group. The assessment and the rights of all participants were protected. A Clinical Research Ethics Committee approved the study's experimental procedures (registration number NCT04939181).

### 2.3. Participants

Twelve seasoned athletes participated in this investigation (mean $\pm$ SD, age: $20.6 \pm 2.7$ year; height: $197.8 \pm 11.7$ cm; and body mass: $89.0 \pm 21.2$ kg). These skilled individuals were members of a distinguished Spanish Euroleague squad, showcasing their talents in the competitive arena of LEB Plata, Spain's third division basketball league. The trial inclusion criteria were: (1) participants did not present any injury or pathology during the study; (2) subjects were required to be in a normal condition, mentally competent to participate in the study and in a condition to complete the study questionnaires. The exclusion criteria were: (1) presenting contraindications for treatment with NESA noninvasive

neuromodulation such as a pacemaker, internal bleeding, with skin in poor condition, such as ulcerations or wounds, that would prevent application of electrodes, acute febrile processes, acute thrombophlebitis and/or electricity phobia; (2) players who did not sign the informed consent; (3) presenting any injury or pathology during the study. The entire team was included in the study and volunteers were randomized into groups. This study was approved by the Clinical Research Ethics Committee of the Catalan Sports Council (Government of Catalonia) following the Helsinki declaration for ethical considerations, with number 006/CEICGC/2021 (Clinicaltrial.gov registration number NCT04939181 on 25 June 2023).

*2.4. Randomization and Groups*

Players who consented to participate were randomly assigned to one of the two treatment groups (real or placebo device). A fixed-size block design, generated by the data manager, was employed to ensure balanced randomization across both arms. The allocation process was conducted with careful impartiality by an unbiased investigator within the designated research team (refer to Figure 1).

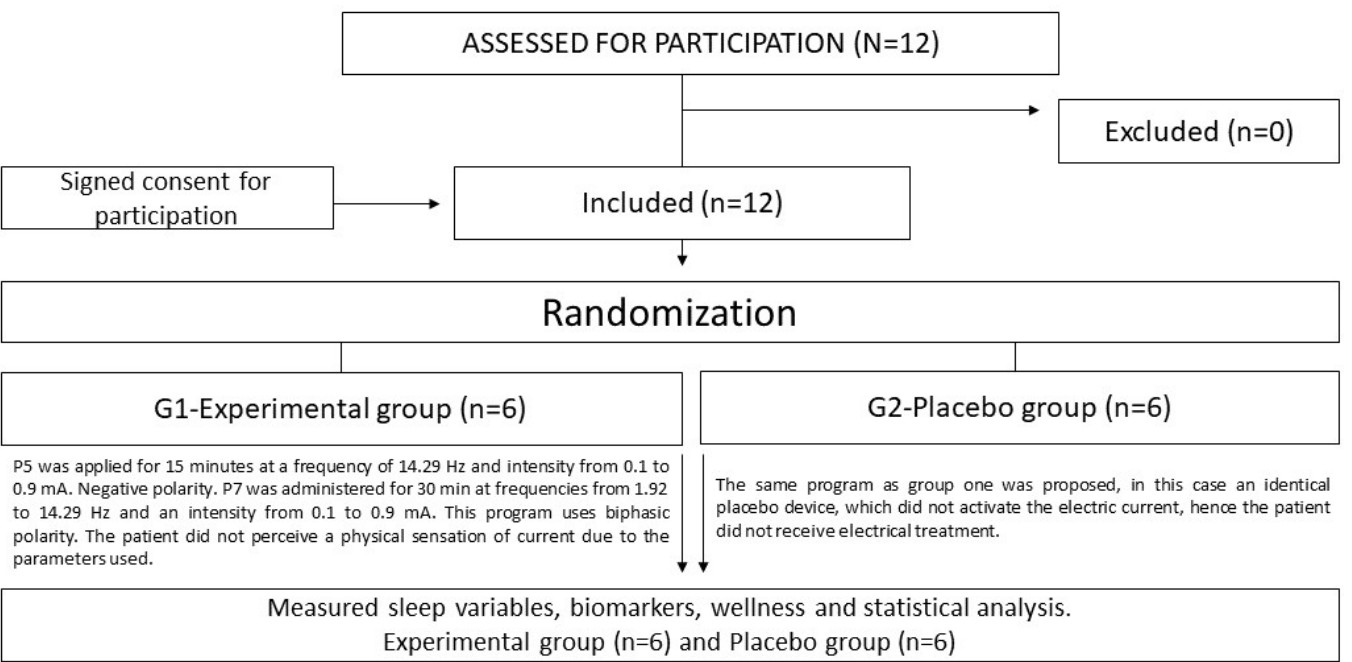

**Figure 1.** Flowchart of the intervention process.

*2.5. Intervention*

Engaging in weekly bouts, the team embarked on one to two matches, following a customary 45 min warm-up routine comprising dynamic stretching. Subsequently, they performed tailored mobility drills and honed individual basketball prowess through targeted skills practice encompassing shooting, passing, and dribbling. The matches adhered strictly to the regulations set forth by the International Basketball Federation (FIBA). Moreover, the team adhered to a team sports-oriented regimen known as "structured training", pioneered by FC Barcelona, aimed at priming athletes for competitive team sport engagements [19]. This methodology revolves around two distinct training modalities: coadjutant sessions, encompassing general off-court exercises like split squats and single-arm presses, and optimizing sessions, focusing on sport-specific on-court drills such as small-sided games and full-scale 5-on-5 scrimmages [20,21]. Typically, the team observed a day of rest following each match within their microcycles (the period between official games or matches). However, during the fourth week, the team transitioned into playoffs—a climactic phase involving a final contest or series of contests to determine the victor among tied contestants or teams [22].

All players underwent a NESA protocol twice weekly, typically between noon and 2 pm, following Tuesday and Thursday training sessions and preceding lunch. During the 45 min intervention, all 12 players were connected to NESA units; however, only half of the devices were appropriately configured, with the other half reserved for the placebo group. Throughout the duration of connection, players remained seated or reclined on mats.

A triple-blind capture system was used (neither the specialist nor anyone in charge knew which players were entering the complementary treatment), as well as two NESA XSIGNAL® devices (NESA noninvasive neuromodulation) in double-blind mode.

NESA technology operates through coordinated electrical stimulation utilizing 24 semi-electrodes, effectively modulating the autonomic nervous system via low-frequency electric signals. This stimulation targets various areas of the body by means of circulating current, all while remaining minimally invasive, applied solely to the body's surface. The current characteristics include emission of low-frequency pulses ranging from 1.3 Hz to 14.29 Hz, with pulse intensity varying between 0.1 and 0.9 milliamps and a potential difference of $\pm 3$ V. With coordination across 24 electrodes (six electrodes per limb) simultaneously stimulated, the resultant effect is systemic rather than localized to a specific muscle or nerve area, owing to the distributed nature of the electrode placement and the electrical parameters involved (Figure 2) [16].

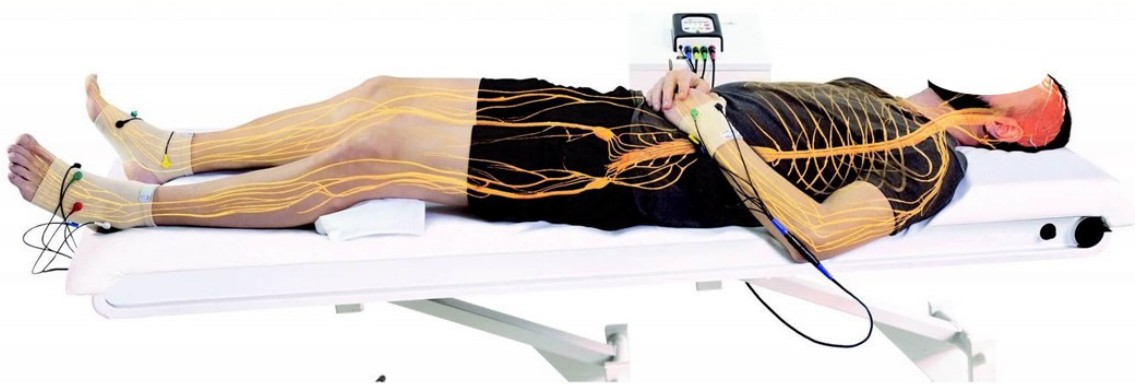

**Figure 2.** A diagram representing the electrodes located on wrist and ankles. The NESA device can be seen in the background.

According to the proposed randomization, the sample was divided into two groups (Figure 1). In the experimental group (G1), NESA was applied after the normal training session twice a week for a period of 6 weeks. The administered protocol comprised Program 5 (P5) and Program 7 (P7). Initially, P5 was implemented for a duration of 15 min, operating at a frequency of 14.29 Hz and an intensity ranging from 0.1 to 0.9 mA. Notably, the polarity remained negative throughout the application of this program. Subsequently, P7 was employed for a period of 30 min, featuring frequencies spanning from 1.92 to 14.29 Hz and an intensity within the range of 0.1 to 0.9 mA. The individual did not experience any physical sensation of current owing to the parameters employed. Furthermore, a placebo group (G2) was introduced, wherein a program akin to the intervention group was administered. However, in this case, an identical placebo device was utilized, which did not initiate an electric current; thus, the subjects in this group did not receive any electrical treatment. In the experimental and placebo groups, the wrist and ankle electrodes were fitted after the skin had been cleaned with an alcohol solution. In addition, the directional electrode was placed in central mode, located near the C7 area (between vertebrae)

### 2.6. Screening and Outcome Measures

*Sleep parameters*: To conduct the sleep assessment, participants were provided with an Oura ring actigraph [23] (Oura Health, Oulu, Finland), which they had to wear on one finger 24 h a day (except during training sessions and matches). At the beginning of the study, each participant had to download the Oura application from their mobile

phones and created an Oura account. Participants were instructed to launch the application each morning to upload data from the ring to the application interface. Uploaded data were automatically transferred (via internet connection) to the study database in the Oura cloud service. The Oura ring measures sleep and recovery variables based on resting heart rate (HRrest), heart rate variability (Rmssd), and motion using photoplethysmography (a variable used to determine circulatory capacity through the heart rate variability in actigraphy) and an accelerometer. The Oura ring classifies the records into three phases: light sleep (sleep phases 1 and 2 by seconds), deep sleep (sleep phases 3 and 4 by seconds) and REM (rapid eye movement) sleep (REM sleep by seconds), and a combination of them with the variables: duration (minutes of the light, deep and REM phases), total (the sum of sleep duration and awake phases), and awake (seconds spent at stage 0). Previous research had a 57% accuracy in sleep phase classification. This study chose the following variables: duration, total, awake, light sleep, deep sleep, and REM [23–25]. The actigraphy method is inexpensive and can be administered with minimal impact on habitual sleep or training routines. As such, actigraphy is the preferred method for objectively monitoring the sleep of athletes [26].

*Biochemical parameters*: Peripheral blood is the concentration of cortisol, testosterone, and testosterone:cortisol ratio [27,28]. The following concentrations are proposed to analyze the degree of post-match cellular homeostasis imbalance in semi-professional players to observe possible significant differences between pre- and post-match values (48 h pre-match, immediately after match, and 48 h after match, following the microcycles). Cortisol concentration defines the level of stress that a player is submitted to during the microcycle, as a catabolic hormone that promotes the mobilization of energy substrates and is responsible for suppressing immune system function. For its part, testosterone concentration helps to describe recovery processes and player adaptation after the stress produced by a training session or match, as a hormone engaged in multiple physiological functions, particularly in charge of the increase and maintenance of skeletal muscle, bones, and RBCs. Additionally, the T:C ratio delineates the equilibrium between catabolic (cortisol) and anabolic (testosterone) processes [10,29]. The analyses were performed on saliva samples using ELISA assays conducted by DRG Instruments GmbH. Both assays were performed using solid-phase enzyme-linked immunosorbent assay (ELISA), based on the competitive binding principle on ELISA Triturus Analyzers (Grifols, Barcelona, Spain). The testosterone:cortisol ratio was determined by dividing the testosterone value by the cortisol value.

### 2.7. Statistical Analysis

Statistical data analysis was conducted using IBM SPSS Statistics for Windows, Version 27.0 (Armonk, NY, USA: IBM Corp). Categorical variables were summarized using percentages and relative frequencies, while numerical variables were summarized using the mean and standard deviation (SD). To assess the normality of the samples, the Kolmogorov–Smirnov and Shapiro–Wilk tests were utilized. For comparing the means of two independent samples, either the Student's *t*-test or the non-parametric Mann–Whitney U test was employed. Pearson's correlation coefficient was used to evaluate the relationship between two numerical variables. The results were statistically significant if *p*-value was <0.05. Because this was a preliminary study and many of the sleep measurements were taken daily, we decided to perform a repeated sample analysis, in the interest of seeing how the group, and each player, evolved over the weeks.

### 3. Results

### 3.1. Sample

A total of 12 participants (mean ± SD, age: 20.6 ± 2.7 year; height: 197.8 ± 11.7 cm; and body mass index: 89.0 ± 21.2 kg) met the inclusion criteria and were randomized into the data collection process.

### 3.2. Sleep

For each group, all the observations were recorded for six weeks. In the intervention group, $n = 240$ observations (repeated measures for 48 days) were collected for six athletes who used NESA. The number of observations recorded for each player was not uniform, due to the Oura data being recorded daily. A similar situation was evident with the placebo group, with $n = 228$ observations for the six weeks. For the categories awake, light, and deep, we did not find significance differences between groups with repeated measurement; but we determined significance differences for other Oura variables such as REM sleep, duration, and total (wee the significant variables summary in Table 1 represented by minutes).

**Table 1.** Significant variables summary described by seconds, minutes, and hours.

| Variable | Group | Mean (SD) | Median (IQR) | Min/Max | *p*-Value |
|---|---|---|---|---|---|
| DURATION (minutes) | NESA ($n = 240$) | 464.16 (88.98) | 464.5 (109) | 263/816 | <0.001 * |
|  | PLACEBO ($n = 228$) | 432.23 (89.72) | 432.5 (119.75) | 227/728 |  |
| REM (minutes) | NESA ($n = 240$) | 50.98 (34.1) | 52 (58.5) | 0/137.5 | 0.007 ♣ |
|  | PLACEBO ($n = 228$) | 42.44 (26.66) | 37.5 (37.38) | 0/118 |  |
| TOTAL (minutes) | NESA ($n = 240$) | 378.9 (73.65) | 384.25 (102.25) | 200.5/649 | <0.001 * |
|  | PLACEBO ($n = 228$) | 346.88 (66.35) | 344.5 (91.13) | 183/537.5 |  |
| AWAKE (minutes) | NESA ($n = 240$) | 85.25 (42.68) | 77.75 (41.38) | 24/281.5 | 0.442 ♣ |
|  | PLACEBO ($n = 228$) | 91.99 (38.11) | 78.25 (47.64) | 15.5/290 |  |
| DEEP (minutes) | NESA ($n = 240$) | 102.83 (38.95) | 103 (67.75) | 49/153 | 0.104 ♣ |
|  | PLACEBO ($n = 228$) | 93.83 (45.4) | 96 (61.5) | 39/170 |  |
| LIGHT (minutes) | NESA ($n = 240$) | 263.33 (54.96) | 246.2 (869.63) | 217.5/349 | 0.096 ♣ |
|  | PLACEBO ($n = 228$) | 198.25 (52.96) | 193.1 (99.30) | 141/280.5 |  |

SD = standard deviation; IQR = interquartile range = Percertile75-Percentile25; ♣ Non parametric Mann–Whitney U test; * Student t–test.

For duration, a significant difference of 31.93 min was detected in the experimental group comparing mean durations between the NESA and PLACEBO groups (*p*-value < 0.001; t = 3.865 df = 466). For the REM variable, we found a significant difference of 14.5 min in favor of the experimental group comparing the mean values of the NESA group and the PLACEBO group (*p*-value = 0.007; U = 23424). Finally, the total variable in the intervention group showed a significant difference of 32.02 min more than the placebo group mean (*p*-value < 0.001; t = 4.931 df = 466).

The following graphs correspond to the interpolation lines of the observations of each group. The horizontal line represents the average sleep duration value (Figure 3) of the experimental group (7.74 h). In this case, the mean duration of sleep in the NESA group is seen to be higher than that of the placebo group. The difference was 31.93 min. The duration variable in the samples of both groups met the normality hypothesis (Kolmogorov–Smirnov test $p > 0.05$, $p < 0.001$), considering the equality of variances ($p > 0.05$). In this case, the mean duration of sleep in the NESA group is seen to be higher than that of the placebo group. The difference was 31.93 min.

The comparison shows that the NESA group had a higher REM time during sleep duration ($p = 0.007$) than the placebo group. Therefore, the experimental group had a longer REM phase duration (8.54 min) than the placebo group. Moreover, the most notable discrepancy was observed over weeks four and six, when the team began the playoffs (Figure 4).

The total sleep time variable showed differences between both groups ($p < 0.001$); the mean was higher in the experimental group (difference of 32 min). The red and blue forms correspond to the interpolation lines of the observations of each group. The horizontal line corresponds to the value of the average total sleep time of the experimental group (278.90 min/6.32 h) (Figure 5).

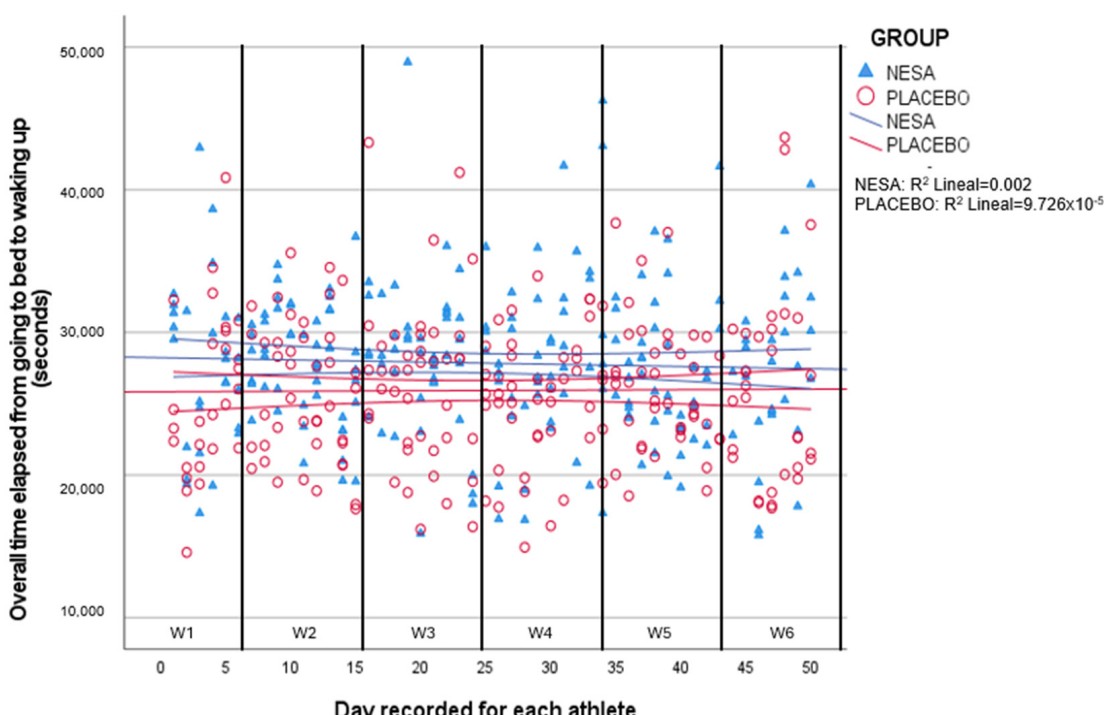

**Figure 3.** Scatter diagram of the duration variable on each day recorded for each athlete.

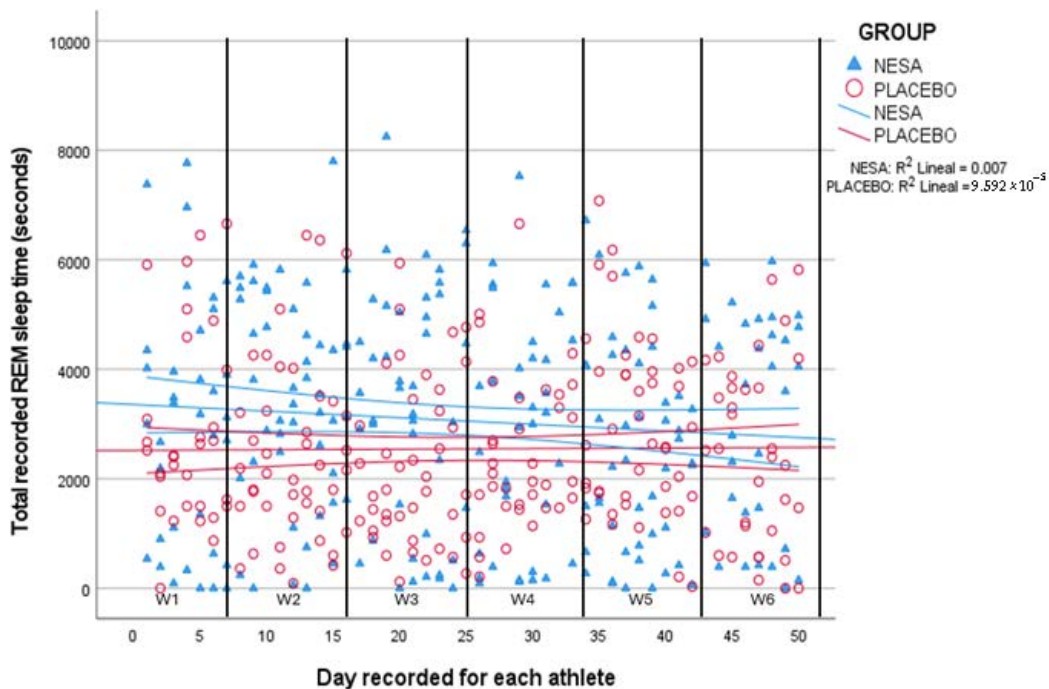

**Figure 4.** Scatter diagram of the REM variable on each day recorded for each athlete.

*3.3. Biochemical Parameters*

The analysis of the cortisol and testosterone hormones and the ratio between them, T:C, over the six-week intervention, presented differences in means between groups for the cortisol variable ($p = 0.045$) but not for the testosterone and T:C ratio variables.

Both groups showed similar variations in cortisol levels over the six weeks. We would like to highlight the lower baseline level of the placebo group (3.5 nmol/L) in comparison to the intervention group (4.81 nmol/L) (Table 2).

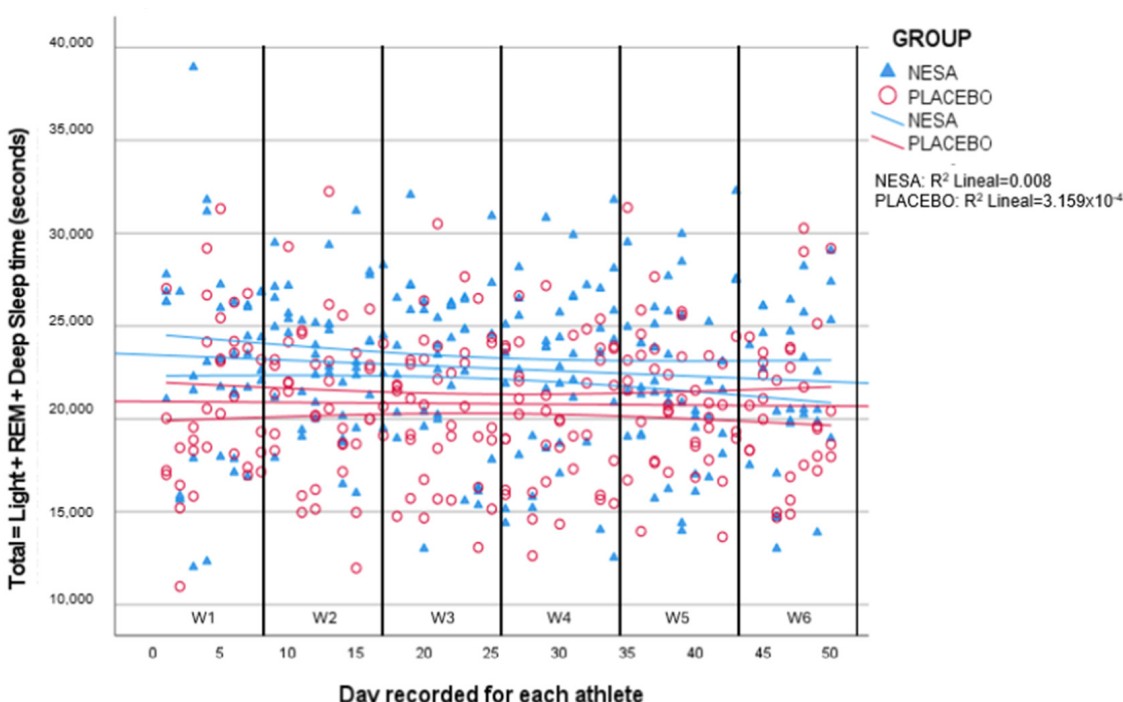

**Figure 5.** Scatter diagram of the total variable on each day recorded for each athlete.

**Table 2.** Descriptive table of the biochemical parameters such as cortisol (C), testosterone (T) and testosterone:cortisol ratio.

| GROUP | VARIABLE | Mean (SD) | | | | | |
|---|---|---|---|---|---|---|---|
| | | WEEK 1 | WEEK 2 | WEEK 3 | WEEK 4 | WEEK 5 | WEEK 6 |
| NESA | T:C | 25.95 (11.90) | 23.04 (16.21) | 35.95 (30.85) | 17.55 (19.64) | 8.10 (4.45) | 27.71 (13.62) |
| | C | 4.81 (3.85) | 5.60 (4.96) | 4.23 (2.74) | 4.75 (3.18) | 4.57 (2.72) | 3.66 (1.99) |
| | T | 96.48 (30.25) | 94.27 (52.68) | 112.07 (47.47) | 61.18 (53.36) | 33.38 (20.36) | 90.17 (50.23) |
| PLACEBO | T:C | 37.08 (22.87) | 29.68 (14.92) | 41.19 (22.64) | 30.42 (39.84) | 10.98 (7.89) | 44.02 (24.43) |
| | C | 3.50 (1.67) | 3.93 (1.41) | 3.28 (1.20) | 4.36 (2.18) | 4.48 (2.57) | 3.46 (1.68) |
| | T | 103.69 (33.34) | 107.17 (45.36) | 115.65 (33.63) | 85.85 (59.25) | 42.92 (31.52) | 133.54 (56.18) |

Regarding cortisol and considering the baseline values of each group due to interpersonal variability, the distribution between the two groups over the microcycles was very similar. Cortisol values remained constant in the intervention group, while a non-significant increase was observed in the placebo group in weeks four and five. Similarly, significant differences were found in the mean between the intervention group and the placebo group at six weeks ($p = 0.45$) (Figure 6).

No significant differences were observed between groups for the testosterone variable; however, weekly analysis presented a similar distribution between groups. Although it was greater in the placebo group, a downward trend was observed in both groups. Weeks four and five presented a reduction in testosterone levels in both groups too (Figure 7).

In the T:C ratio analysis, the study showed no significative differences between the groups, nevertheless we observed similar distributions in both groups in weeks four and five. Although no significant differences were obtained, it should be noted that testosterone values, and therefore the testosterone: cortisol ratio, decreased in weeks four and week five in both groups.

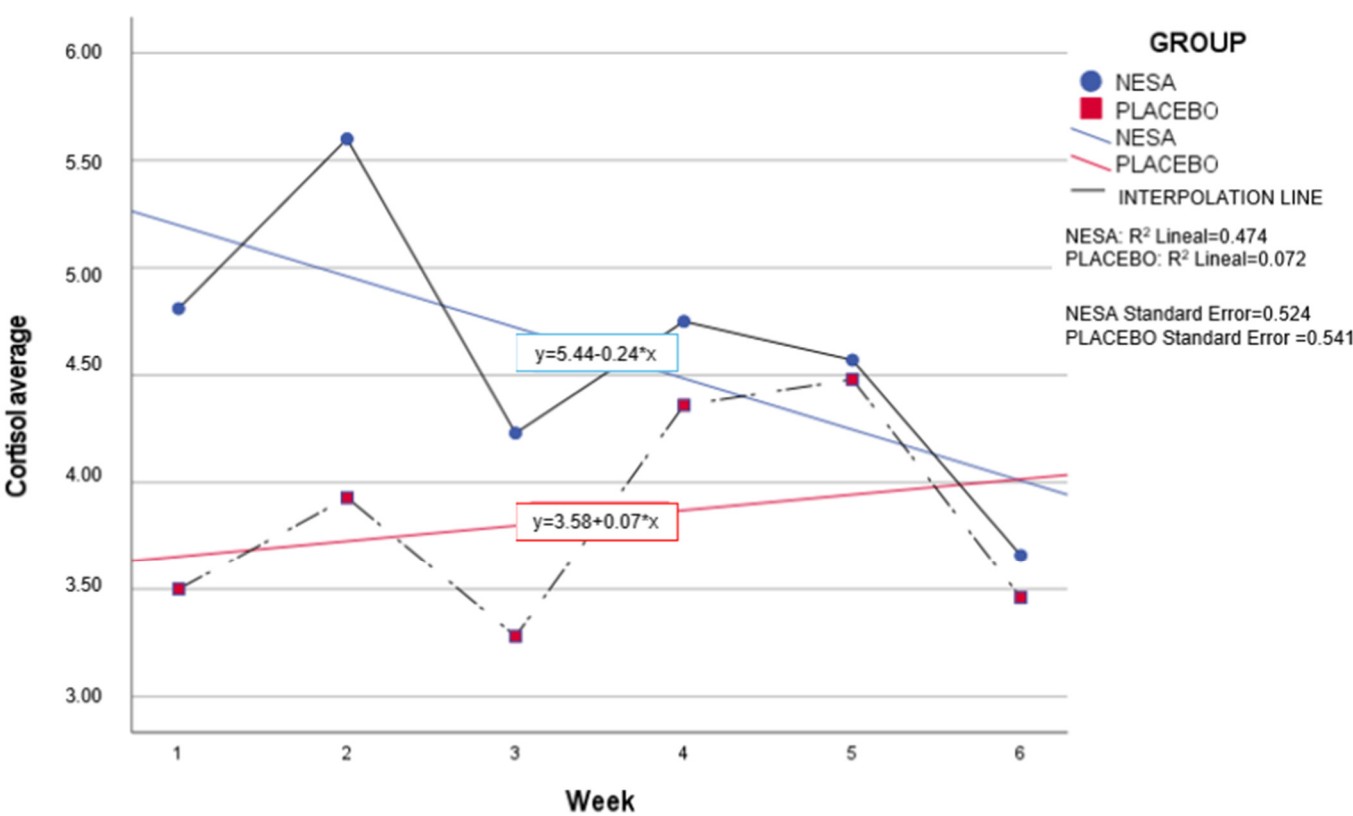

**Figure 6.** Distribution of salivary cortisol concentration by group throughout the weeks of treatment. The table is complemented with lines that plot the concentration trend in each group.

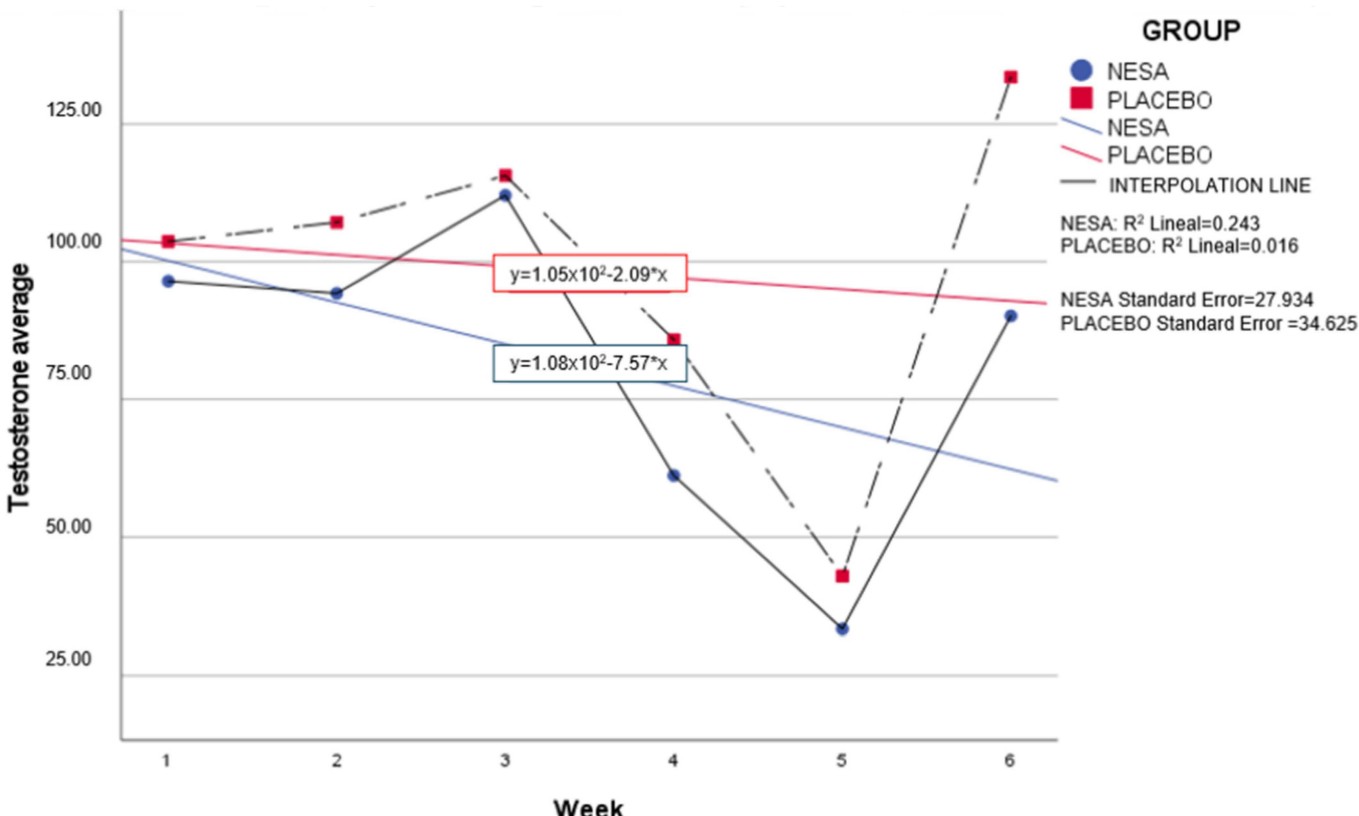

**Figure 7.** Distribution of testosterone means concentrations by groups over the six weeks of treatment. The table includes lines plotting the concentration trend in each group.

## 4. Discussion

The results found in this study in the significant sleep variables, REM, duration, and total, showed that the intervention group maintained and even improved these variables over the six weeks, particularly from the fourth week onwards (when the team started the playoffs). Althoug phases such us light and deep phases were not significant, they also improved in the experimental group. It is also known that sleep physiology is modulated by the autonomic nerve system so that NESA noninvasive neuromodulation, as an autonomic modulator for parasympathetic and sympathetic balance based on microcurrents, can be a useful electrotherapy with the ability to produce changes in sleep and autonomic systems [16].

The availability of a noninvasive neuromodulation device such as NESA technology to a professional team opens the possibility of its integration into their daily training regimen. Furthermore, it serves as a convenient and side-effect-free treatment option. By optimizing athletes' autonomic nervous systems, there exists the potential for enhanced future performance and improved sleep quality.

In recent times, elite athletes have experienced challenges with sleep quality due to their demanding schedules and competition requirements. Therefore, enhancing sleep quality could translate into improved performance [4,13]. Good sleep quality and rest have been previously found to influence coordination and muscle recovery [30], as well as the consolidation of memory (procedural) during the REM phase, which is crucial for athletes and players [30–32]. Other techniques for improving sleep quality, such as sleep hygiene education [33] and nutritional intervention [14], have also demonstrated improvements in sleep quality among athletes. However, these techniques rely heavily on the engagement of coaches and trainers and are hindered by limitations associated with being self-directed activities that athletes must fully commit to, a challenge that may be particularly pronounced for young athletes. Integrating NESA neuromodulation techniques as part of daily treatment within sports recovery protocols could promote adherence and offer a conveniently manageable approach for athletes to engage with consistently.

Regarding biomarkers, the cortisol concentration curves were very similar in both groups when considering their respective baseline values. However, the intervention group also exhibited a downward trend, which coincided with the group showing the greatest improvement in sleep variables for duration, REM, and total. This study represents an initial experimental investigation into the use of the NESA neuromodulation technique for sleep and biomarkers. Recent studies conducted by Wu et al. (2008) and Oginska (2020) explored morning serum cortisol concentrations in healthy adult men subjected to sleep restriction to three hours per day. These investigations revealed a noteworthy decrease in morning cortisol levels [34,35].

Previous studies have also investigated the behavior of cortisol and testosterone hormones, as well as the ratio between them, to monitor the recovery status of athletes. This monitoring includes assessing the ability to activate the sympathetic nervous system during physical activity and the capacity to enhance anabolic restorative processes to achieve homeostasis, as indicated by testosterone concentrations. The testosterone:cortisol ratio provides insight into the recovery and fatigue status, as well as the catabolic–anabolic processes that the cellular complexes of players undergo. Additionally, it offers an understanding of the athlete's trend throughout the microcycles or season [6]. Similarly, other studies have focused on examining significant differences before and after a match to assess the short-term sensitivity of these hormones [7,8,36].

The objective of this preliminary study was to explore differences between the intervention group, which underwent two neuromodulation sessions, to ascertain whether this method enhanced player recovery, and the placebo group, which did not undergo NESA microcurrents. Analysis of hormone levels during the intervention revealed no significant differences between weeks in terms of cortisol, testosterone, and the testosterone:cortisol (T:C) ratio, although a decrease in testosterone and T:C ratio was observed in both groups, likely attributable to the stress and mental fatigue experienced in weeks four and five.

The team's strength and conditioning coach provided a contextual log for the intervention weeks. The week four post-match period began with a two-day lockdown of the players due to COVID-19, followed by PCR testing and a midweek home match on Wednesday without any training sessions. They played another home match on Sunday, resulting in a loss. Week five was finals week, during which the team trained for five consecutive days. The coaching staff noted a general sense of stress within the team atmosphere, compounded by travel of more than seven hours by bus, culminating in a loss in the final match on Sunday. Previous studies have reported an association between basketball players' mood and testosterone and cortisol concentrations before and after a game [12]. This finding may elucidate the decline in testosterone levels in both groups, attributed to the observed anxious and stressful environment noted by the coaching staff.

Arruda et al. [7] also observed significant differences in testosterone concentration before training, with lower concentrations observed compared to before a competitive match. Similarly, higher pre- and post-match cortisol levels were observed in games as particularly challenging in medium-difficulty matches. These findings align with the elevated cortisol levels found in the placebo group, while the intervention group exhibited maintenance of these levels. Future research could potentially investigate the use of alternating microcurrents to help maintain cortisol values during critical weeks, such as those during a professional basketball team's playoff.

Our research is a preliminary study conducted using NESA noninvasive neuromodulation in the elite sports setting. However, there are some limitations to be reported: (1) a small sample size collected, as the data were collected from a single basketball team (2); Oura is a consumer-grade, not research-grade, device and it is convenient but not highly accurate; (3) the lack of ANS activity analysis to understand better the sleep-stage-dependent influence of NESA. Therefore, further studies with a larger sample size in other sports are required, and comparisons should be drawn with coordination, strength, and HRV aspects to obtain a better understanding of the relationship between basketballers' performance and sleep quality improvement. Future research should seek to fine-tune biomarker variables and replicate these methods in follow-up studies.

## 5. Conclusions

NESA noninvasive neuromodulation appears to be an effective treatment for improving sleep quality, specifically in the REM phase, duration, and total duration during the night in elite athletes. This could potentially open an interesting research avenue in the neuromodulation of the autonomic nervous system and its relationship with sleep quality as an integral part of daily recovery techniques in elite sports. Regarding biomarkers, significant differences were observed for cortisol between the intervention and placebo groups. However, further studies are warranted to confirm the causal relationship between cortisol levels and NESA noninvasive neuromodulation. It is imperative to exercise caution in interpreting these findings given the preliminary nature of this study. Nevertheless, in conclusion, NESA noninvasive neuromodulation presents a promising approach for easily and superficially modulating the autonomic endogenous function and improving sleep quality.

**Author Contributions:** Conceptualization, M.M.S.; data curation, F.G.; formal analysis, M.d.P.Q.-M.; investigation, F.G., F.P. and D.D.Á.-A.; methodology, M.M.S. and F.P.; project administration, M.M.S. and F.P.; resources, A.B.-S.; software, F.G. and M.d.P.Q.-M.; supervision, R.M.-R. and D.D.Á.-A.; validation, M.d.P.Q.-M.; visualization, A.B.-S., E.T.H. and D.D.Á.-A.; writing—original draft, R.M.-R.; writing—review and editing, R.M.-R. All authors have read and agreed to the published version of the manuscript.

**Funding:** This research received no external funding.

**Data Availability Statement:** The protocol of the study can be requested via email to estherteruel.hernandez@gmail.com.

**Acknowledgments:** We would like to thank the 12 professional basketball players who participated in the study, as well as the coaching staff for giving us the opportunity to carry out this study.

**Conflicts of Interest:** The authors declare no conflict of interest.

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
