# Peer review of "Effects in Sleep and Recovery Processes of NESA Neuromodulation Technique Application in Young Professional Basketball Players: A Preliminary Study"

_stresses, doi:10.3390/stresses4020014_

Round 1

Reviewer 1 Report

Comments and Suggestions for Authors

1. Please rephrase those large parts that are marked by ithenticate.

2. Part 2.6: "plethysmography"? I do not know what the authors meant, but surely not this.

3. Part 3.2: "yhe6"?

4. Data should be presented either as mean or median depending on the data distribution

5. Please clearly state limitations of the study.

Comments on the Quality of English Language

Some corrections required

Author Response

We would like to notice both reviewers that the paper have been revised by a native speaker and some words and sentences have been modified. You will find these modifications in the text by blue mark. Thank you for your suggestion, we believe that we have increased the paper´s quality thanks your advice.

1.Please rephrase those large parts that are marked by ithenticate.

 Sorry but cannot identified the intention of this suggestion. In the version download did not appear any marks or colors “by ithenticate”. Sorry but we do not know what is “ithenticate” system. Could you please explain the suggestion?. Anyway, we have revised all the document in terms of English and grammar.

2.Part 2.6: "plethysmography"? I do not know what the authors meant, but surely not this.

The plesthysmography exist and it is a kind of measurement use in actigraphy, such us OURA ring, based on volume changes and heart rate variability. You will find more information in the references below. Anyway, thank you for your suggestion because we have included in the text a brief explanation for the readers.

Z Beattie et al 2017 Physiol. Meas. 38 1968 DOI 10.1088/1361-6579/aa9047

Pedro Fonseca, Tim Weysen, Maaike S. Goelema, Els I.S. Møst, Mustafa Radha, Charlotte Lunsingh Scheurleer, Leonie van den Heuvel, Ronald M. Aarts, Validation of Photoplethysmography-Based Sleep Staging Compared With Polysomnography in Healthy Middle-Aged Adults, Sleep, Volume 40, Issue 7, July 2017, zsx097, https://doi.org/10.1093/sleep/zsx097

3.Part 3.2: "yhe6"?

This was a typo mistake. We have corrected and marked in yellow.

4.Data should be presented either as mean or median depending on the data distribution.

Thank you for the suggestion, we have realized that we used, by error, “median” term in page 7. We have corrected and marked in yellow.

5.Please clearly state limitations of the study.

Thank you for your advice. We have included the limitation in page 12 and 13. (see in yellow)

Thank you for your recommendations. The attached document contains all the corrections based on the suggestions of both reviewers. 

Reviewer 2 Report

Comments and Suggestions for Authors

Methods

Explain the meaning of the metrics provided by Oura. TOTAL refers to total sleep time which is equal to sleep duration minus wake-after-sleep-onset

Results

Table 1 should report the values of all metrics (not only the ones that were significantly different between conditions.

If total sleep time increases in the Stim condition and so does REM but to a lesser extent, the rest of the increasing must be coming from Light or Deep NREM. 

Figure 3, 4, and 5 would be far more clear if aggregated in weeks in a similar manner as it was done in Figures 6 and 7. After all, the discussion reports the outcomes in weeks not in days. 

Figures 6 and 7 please add the standard error bars to better understand the significance of trends.

Figure 6. What explains the baseline difference in cortisol concentration between the NESA and Placebo groups? 

Discussion

Discuss the implications of sleep duration increase and REM significance. What happens for NREM (no change?). If so, what are the hypotheses about the selective influence of NESA on REM sleep.

The first paragraph in the Discussion section suggests that NESA may produce changes in sleep by means of its influence on autonomic nervous system activity (ANS). It is an interesting hypothesis. Oura provides HR/HRV metrics. Could those be analyzed to provide at least some indication on whether ANS activity is modified by NESA? Given the easiness with which HR/HRV can be obtaining in Oura cloud, this should be an easy but relevant analysis to do.

In the 3rd paragraph, REM is mentioned as a promoter of memory consolidation. Please add the specific type of memory that the authors refer to (procedural, declarative, implicit, etc)

Limitations

Mention the limitation of Oura as being a consumer-grade not research-grade device. it is convenient but not that accurate. 

Are the authors considering a cross-over research follow-up in which the groups receive both NESA and Placebo in a random order? That could be a powerful design to test NESA

In the data availability statement, the idea is to give access to the data used in the analysis not the protocol. In principle, the manuscript describes the protocol in sufficient detail to understand the research design and methods. Please state whether or not the data can be made available on request.

Comments on the Quality of English Language

There are several typos in the manuscript. It seems it can be a problem with format conversion or keyboard mapping. Make sure to proof-read and correct those.

Author Response

We would like to notice both reviewers that the paper have been revised by a native speaker and some words and sentences have been modified. You will find these modifications in the text by blue mark. Thank you for your suggestion, we believe that we have increased the paper quality thanks your advice.

Methods

Explain the meaning of the metrics provided by Oura. TOTAL refers to total sleep time which is equal to sleep duration minus wake-after-sleep-onset. Thank you. We have realized that this part was a bit confusing, so we have rewritten, and we hope you will find correct.

Results

Table 1 should report the values of all metrics (not only the ones that were significantly different between conditions.  Thanl you for your suggestion. We have modified the table and included the rest of variables. See in page 7 in yellow.

If total sleep time increases in the Stim condition and so does REM but to a lesser extent, the rest of the increasing must be coming from Light or Deep NREM. Actually, these NREM variables, light and deep, increase but are not significant. We appreciate your advice and have included this statement in the discussion.

Figure 3, 4, and 5 would be far more clear if aggregated in weeks in a similar manner as it was done in Figures 6 and 7. After all, the discussion reports the outcomes in weeks not in days. Thank you for the suggestion, we have included the weeks in the graphs, it would be better for readers. Thank you.

Figures 6 and 7 please add the standard error bars to better understand the significance of trends. Thank you. We have included the data in the new graphs 6 and 7.

Figure 6. What explains the baseline difference in cortisol concentration between the NESA and Placebo groups? Thank you for your appreciation. The reality is that the cortisol is a hormons with high variavility betweens subjets. In studies with cortisol the variability between people and patiens are realy huge, in fact it is complicated to stabish cortisol standard concetration in population because it can be alterated in shor periods of time due to the stress, the thinking the situation and others emotional condition. We have clarified this concept in the text and also added a new reference about cortisol special condition. We highly appreciate your question, it could help the reader a better compression of the paper.  

Discussion

Discuss the implications of sleep duration increase and REM significance. What happens for NREM (no change?). If so, what are the hypotheses about the selective influence of NESA on REM sleep. Thank you for your recommendation. We have included in the first paragraph of the discussion the effect it has had on both REM and NREM (deep and light) phases. It is described that although the changes in NREM are not significant, they do increase in the post analysis. So NESA can improve the quality of sleep in all phases even though in this case only REM, DURATION AND TOTAL were significant. We add the possible hypothesis of the benefit that can be attributed to NESA.

The first paragraph in the Discussion section suggests that NESA may produce changes in sleep by means of its influence on autonomic nervous system activity (ANS). It is an interesting hypothesis. Oura provides HR/HRV metrics. Could those be analyzed to provide at least some indication on whether ANS activity is modified by NESA? Given the easiness with which HR/HRV can be obtaining in Oura cloud, this should be an easy but relevant analysis to do.

Thank you for your suggestion. In this case it was not analysed, due to our interest in a preliminary sleep study. It could be very interesting for the new extension study we are planning to analyse in a more exhaustive way the changes in HRV according to age and activity. Thank you for your suggestion and we will take it into account.

In the 3rd paragraph, REM is mentioned as a promoter of memory consolidation. Please add the specific type of memory that the authors refer to (procedural, declarative, implicit, etc). En el Thank you for the suggestion, we have clarified about “procedural memory”. You will find in yellow in third paragraph of the discussion.

Limitations

Mention the limitation of Oura as being a consumer-grade not research-grade device. it is convenient but not that accurate. Thank you. We agree with your appreciation. We have included the oura limitations and another that we considered important to mention.  

Are the authors considering a cross-over research follow-up in which the groups receive both NESA and Placebo in a random order? That could be a powerful design to test NESA. Thank you for your recommendation. We are now preparing new research project and a cross-over is one of our targets, if the logistic and the teams permit us to develop it. It is our next step. Thank you, we highly appreciate your advice.

In the data availability statement, the idea is to give access to the data used in the analysis not the protocol. In principle, the manuscript describes the protocol in sufficient detail to understand the research design and methods. Please state whether or not the data can be made available on request. You could find the statement that the data can be available on request on page 13, for sure it could help the readers to understand in a better way. Anyway, thank you for your advice and consideration.

Thank you again for your recommendation. Please do not hesitate to contact us again if further corrections are needed. We believe that the article has better quality with your suggestions.

Round 2

Reviewer 1 Report

Comments and Suggestions for Authors

I know that pletysmography exists. What I mean is that you cannot perform real pletysmography with any kind of hand-held/mobile device. What you mean is Photoplethysmography which is not really related to the pletysmography...

ithenticate report was prepared by MDPI. If it is not available to you directly, please kindly ask the handling editor and change the text accordingly.

Comments on the Quality of English Language

Acceptable

Author Response

Thank you for the clarification of the Photoplethysmography term. We have changed in the text (see in yellow).

We have also revised the English again and rewritten the parts marked in the Ithenticate revision. (see in blue)

I hope you will find the improved final version.

Thank you again for your time and consideration. 

Reviewer 2 Report

Comments and Suggestions for Authors

The authors have addressed all of my comments but the one about the analysis of autonomic nervous system (ANS) activity via HR and HRV from Oura. I understand that this may be done in a follow up study.

Please add as additional limitation to the paper, the lack of ANS activity analysis to understand better the sleep-stage dependent influence of NESA.

Author Response

Thank you for your recommendation and clarification. We have inlcuded in the limitations the lack of HRV analysis. It could help for future replicated studies. 

Thank you again for contributing to increase the quality of the study.

You will find the document attached with the changes. (see in yellow the corrections and in blue the english review)

Round 3

Reviewer 1 Report

Comments and Suggestions for Authors

I am satisfied with the response. The authors have significantly lowered the percentage of text marked by ithenticate. No further suggestions.

Comments on the Quality of English Language

Acceptable